# A systematic review and meta-analysis of yoga for arterial hypertension

Christoph Geiger[1,2]*, Holger Cramer[1,2], Dennis Anheyer[1,2], Gustav Dobos[3], Wiebke Kathrin Kohl-Heckl[3]

1 Institute for General Practice and Interprofessional Care, University Hospital Tübingen, Tübingen, Germany, 2 Bosch Health Campus, Stuttgart, Germany, 3 Department of Internal and Integrative Medicine, Evang. Kliniken Essen-Mitte, Faculty of Medicine, University of Duisburg-Essen, Essen, Germany

* christophwilhelmgeiger@gmail.com

## Abstract

### Background

This systematic review and meta-analysis is an update to prior research to evaluate the effects of yoga for managing prehypertension and hypertension.

### Methods

Medline/PubMed, Scopus and the Cochrane Central Register of Controlled Trials (CENTRAL) were searched from their inception until April 5th 2024. Randomized-controlled trials (RCTs) that compared yoga to any control intervention in participants with diagnosed prehypertension (120–139/80–89 mmHg) or hypertension (≥140/≥90mmHg) were included. Mean differences (MD) and 95% confidence intervals (CI) were calculated. Risk of Bias was assessed using the Cochrane tool.

### Results

30 RCTs with 2283 participants were included. Very low quality of evidence was found for positive effects of yoga on systolic blood pressure (SBP, 26 RCTs, n = 2007; MD = -7.95 mmHg, 95% CI = -10.24 to -5.66, p < 0.01), diastolic blood pressure (DBP, 23 RCTs, n = 1836; MD = -4.93 mmHg, 95% CI = -6.25 to -3.60, p < 0.01) and heart rate (HR, 14 RCTs, n = 1118; MD = -4.43 mmHg, 95% CI = -7.36 to -1.50, p < 0.01) compared to waitlist control. Compared to active control, very low quality of evidence was found for positive effects yoga on SBP (5 RCTs, n = 306; MD = -4.16 mmHg, 95%CI = -10.76 to 2.44, p = 0.22), DBP (5 RCTs, n = 306; MD = -1.88 mmHg, 95%CI = -3.41 to -0.36, p = 0.02) and HR (2 RCT, n = 128; MD = -5.16 mmHg, 95% CI = -8.39 to -1.92, p < 0.01). Overall, the studies showed a high degree of heterogeneity. The effects found were robust against selection, detection and attrition bias.

**Data availability statement:** All relevant data are within the manuscript and its Supporting Information files.

**Funding:** Wiebke Kohl-Heckl is a member of a post-doc scholarship by the Karl und Veronica Carstens Foundation Germany. The work itself did not receive any specific funding. The funders had no role in study design, data collection and analysis, decision to publish, or preparation of the manuscript.

**Competing interests:** All authors declare that no competing interest exist.

## Conclusion

Yoga may be an option for lowering blood pressure in people with prehypertension to hypertension. More and larger high-quality studies are needed to substantiate our findings.

## Introduction

Arterial hypertension (AHT) and related cardiovascular diseases (CVDs) remain a serious public health issue [1]. Today, almost a third of the world's population suffers from AHT and the amount of people affected is constantly increasing [2,3]. Since AHT is often diagnosed with a latency [4], secondary events are common and optimized treatment is necessary. Most risk factors for high blood pressure (BP) are behavioral and environmental [5,6]. Several national and international guidelines base their treatment on lifestyle optimization [7]. Furthermore, medication should be added [8–10], depending on a patients' individual situation and medical history [9,11]. Even though there are no precise recommendations to date, the high co-prevalence of elevated heart rate (HR) among patients with hypertension opens the consideration of HR lowering measures with regard to the connection between increased HR and cardiovascular morbidity and mortality risk [12]. Considering that both BP and HR as being managed by the autonomic nervous system, stress reduction techniques that benefit autonomic regulation could reduce BP and HR equally.

Yoga can support this process by physical activity and stress reduction alike. The stress-reducing effect is most likely due to a regulation of the function of the autonomic nervous system [13]. Therefore, yoga can be a useful addition in AHT therapy. Originally, yoga describes an Indian philosophy that contains a set of physical and spiritual practices. It focuses not only on physical activation, but on a comprehensive lifestyle change [14]. Merely parts of the original idea are implemented internationally. In Western society yoga is mostly recognized as a form of physical exercise with body postures and stretching, supplemented by breathing exercises and/or meditation [14,15]. There are various combinations of these components and the posture exercises can be tailored to suit the abilities of each individual. This can be beneficial for the treatment of patients, but it also makes it difficult to compare different programs. So far, yoga interventions have shown benefits regarding several CVDs/risk factors such as coronary heart disease [16,17], chronic heart failure [18], type-2-diabetes [19,20] and metabolic syndrome [17,21].

With regard to the effect on blood pressure levels, a potential benefit has been shown by several systematic reviews and/or meta-analyses [22–27], although it remains difficult to compare them due to a wide range of different study inclusion criteria. One meta-analysis included randomized controlled trials (RCTs) in which blood pressure measurements were performed, but excluded those with medicated participants [22]. Others included RCTs in which the average blood pressure levels were pre- to hypertensive, but it remained unclear, whether or not all participants were diagnosed with AHT [23]. Moreover, three authors carried out a systematic

review without a meta-analysis [25–27]. So far, only one systematic review and meta-analysis included RCTs with exclusively prehypertensive and hypertensive participants as well as longer program duration with follow-up. Their results were promising, yet inconclusive [24].

Our systematic review and meta-analysis follows the idea of the latter. It includes RCTs, randomized cross-over-studies and cluster-randomized trials. Subgroup analyses were conducted for (i) type of participants (prehypertension vs. hypertension); (ii) type of yoga interventions (including physical postures vs. not including physical postures); (iii) co-medication (allowed vs. not allowed); (iv) sex; (v) ethnicity and (vi) age subgroups. Our aim was to gain a better knowledge of whether yoga can be beneficial in treating AHT and how the effects of different yoga interventions differ.

## Methods

The review was planned and performed according to the Preferred Reporting Items for Systematic Reviews and Meta-Analyses (PRISMA) guidelines [28] and the Cochrane recommendations for systematic reviews [29]. It was registered at PROSPERO under the number CRD42020203937.

### Study eligibility criteria

RCTs, randomized cross-over studies and cluster-randomized trials were eligible if systolic and/or diastolic blood pressure (mmHg) were assessed in adults (aged ≥ 18 years) with diagnosed arterial prehypertension (systolic blood pressure - SBP 120–139 mmHg; diastolic blood pressure - DBP 80–89 mmHg) or hypertension (SBP ≥ 140 mmHg; DBP ≥ 90 mmHg).

Studies were eligible if they compared any form of yoga (except single intervention or no follow up at all) to usual care or any active control intervention. Co-medication was allowed and did not lead to exclusion as long as it did not differ systematically between groups. No language restrictions were applied.

### Search methods

We performed a systematic literature research through the electronic databases Medline/PubMed, Scopus and the Cochrane Central Register of Controlled Trials (CENTRAL), as recommended by Cochrane [29], from their inception until April 5th 2024. The search strategy used for PubMed was as follows: ((Yoga[Mesh] OR yoga[Title/Abstract] OR yogasana[Title/Abstract] OR yogic[Title/Abstract] OR asana[Title/Abstract] OR Pranayama[Title/Abstract] OR Dhyana[Title/Abstract]) AND (Hypertension[Mesh] OR hypertension[Title/Abstract] OR hypertensive[Title/Abstract] OR prehypertension[Title/Abstract] OR Blood Pressure[Mesh] OR blood pressure[Title/Abstract] OR blood pressure determination[MeSH Terms] OR arterial pressure[MeSH Terms] OR systolic[Title/Abstract] OR diastolic[Title/Abstract])). Reference lists of identified original articles or reviews and the tables of contents of the International Journal of Yoga Therapy and the Journal of Yoga & Physical Therapy were searched manually.

### Data extraction and data management

Data extraction was carried out independently by two review authors using an a priori prepared data extraction form. Data on methods (e.g., method of blood pressure assessment), patients (e.g., age, sex, diagnosis, ethnicity), control interventions (e.g., type, frequency, duration), and results were collected. A discussion with a third review author was undertaken with any discrepancies until consensus was reached.

### Risk of bias assessment of individual studies

Two review authors independently assessed risk of bias using the Cochrane risk of bias tool [29]. Following the Cochrane tool, seven given criteria (random sequence generation, allocation concealment, blinding of participants and personnel, blinding of outcome assessment, incomplete outcome data, selective reporting, and other bias) were assessed and rated

(low, unclear, or high risk of bias). Once more, discrepancies were discussed with a third reviewer until consensus was reached.

## Data analysis

### Assessment of overall effect size

Effects of yoga compared with different control interventions were analyzed separately. Meta-analyses were conducted using a random effects model if at least 2 studies assessing this specific outcome were available. The software used for statistical analyses was Review Manager 5 (version 5.4; The Nordic Cochrane Center, Copenhagen, Denmark). Mean differences (MDs) between groups and their 95% confidence intervals (CIs) were calculated from means, standard deviations (SDs), and group sizes using the inverse-variance method of meta-analysis [30]. For funnel plots and meta-regression R software Version 4.1.1 (R Foundation for Statistical Computing, Vienna, Austria. URL https://cran.r-project.org) and the 'meta' package [31] were used.

### Assessment of heterogeneity

Heterogeneity was analyzed using $I^2$ statistics with heterogeneity categorized as (i) $I^2 = 0\%–24\%$: low heterogeneity; (ii) $I^2 = 25\%–49\%$: moderate heterogeneity; $I^2 = 50\%–74\%$: substantial heterogeneity; and (iii) $I^2 = 75\%–100\%$: considerable heterogeneity [29,30]. The $\chi^2$ test was applied to measure whether differences in results are compatible with chance alone; $p \leq 0.10$ was considered to indicate significant heterogeneity. To explore the underlying between-study variability we estimated $\tau^2$ as an additional means. Unlike the $I^2$ statistics, $\tau^2$ does not systematically increase with the number of studies or the sample size.

### Subgroup analyses

Subgroup analyses were performed for (i) participant group (prehypertension vs. mixed sample vs. hypertension), (ii) intervention type (physical postures vs. no physical postures), (iii) co-medication (allowed vs. not allowed), (iv) type of BP measurement (ambulatory vs. clinical) and (v) type of active control (exercise vs. health education).

### Sensitivity analyses

To test the solidity of significant results, we conducted sensitivity analyses for studies with high or unclear vs. low risk of bias at the domains selection bias, detection bias, and attrition bias. In case statistical heterogeneity was present in the respective meta-analysis, subgroup and sensitivity analyses were also employed to investigate potential causes for heterogeneity.

### Risk of bias across studies

Publication bias was assessed for SBP, DBP and HR in studies comparing yoga to usual care using funnel plots, checking for visual distribution and asymmetry and Egger's Test.[32]

### Moderator analyses

We performed an univariate meta-regression for SBP, DBP and HR in studies comparing yoga to usual care by total duration time of yoga interventions.

### Quality of evidence

The quality of evidence for each outcome was assessed as high quality, moderate quality, low quality, or very low quality according to the Grading of Recommendations Assessment, Development and Evaluation (GRADE) approach [33], based on the methodological quality and the confidence in the results.

## Results

### Literature search and study selection

The literature search yielded 3131 records. After screening of non-duplicate article titles and abstracts, 2135 records were excluded because they were not randomized, participants were not diagnosed with (pre-)hypertension, and/or yoga was not an intervention. 71 full-text articles were assessed for eligibility. Among them, 12 studies were not randomized. One duplicate article was removed and for one record there was just a conference abstract available. Three RCTs also included non-hypertensive participants. Two more were excluded because though hypertensive participants were included, BP measurements were not monitored as an outcome. Twelve more articles were excluded for multimodal interventions or not using yoga as an intervention at all. Seven RCTs focused on a single session or immediate BP effects and did not perform any follow-up. Three studies were excluded due to their cross-over design causing an overlap of their study cohorts with other included studies. S1 Table provides a summary of the excluded studies and reason for exclusion.

Within 30 full-text articles reporting on 30 RCTs which met the inclusion criteria, 2283 participants were included in the qualitative analysis (Fig 1). Of the 30 included articles, all articles were published in English language.

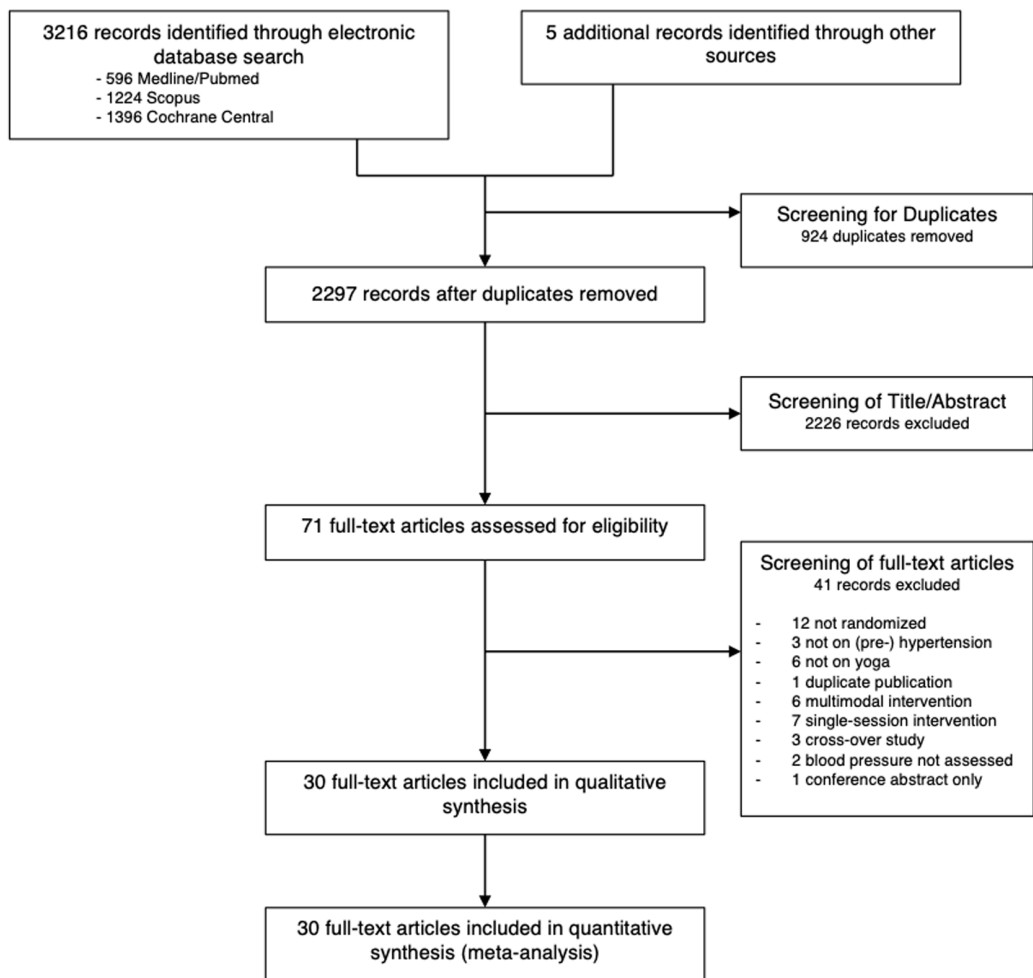

**Fig 1. Flow Chart of the study selection process.**

## Study and participants characteristics

The characteristics of the included studies, intervention characteristics and blood pressure assessment are shown in Table 1. Of the 30 studies, 16 RCTs were originally carried out in India [34–49], one in Indonesia [50], two in Nepal [51,52] and one in Thailand [53]. Four were carried out in the USA [54–57], one in Canada [58] one in Ecuador [59], two in Spain [60,61], one in Germany [62] and one in Sweden [63].

Patients' mean age ranged from 22.5 to 68.9 years, with a median of 50.8 years; 20% to 85.3% of participants were women (median = 52.5%); only three studies reported complete data on ethnicity [54,55,57] with 46,7% to 89% of their participants being white, one study did provide data but did not report the Caucasian proportion [56]. The ethnic distribution of the remaining original articles remains unclear. Therefore, no statement can be made about the average proportion of Caucasians in the total group of participants.

Three studies included prehypertensive participants [34,45], six RCTs had both prehypertensive and hypertensive participants [40–42,54–56], in which we included Shantakumari because of their criterion of hypertensive condition starting from >130/>80mmHg SBP. All other studies included hypertensive participants only.

All RCTs performed yoga interventions in any form, in 22 articles the yoga intervention included physical postures, five studies did not include physical postures in their intervention and focused on pranayama instead [36,42,44,46,57]. Three three-armed RCTs had two intervention groups with one including physical postures and the other one without [59,61,62]. 23 RCTs compared yoga to usual care/waitlist control, four studies carried out active control interventions consisting of exercise [38,56,58] or weekly health education [55], one three-armed study compared yoga to both usual care and exercise [40], two other RCTs compared a multimodal lifestyle-modification plus yoga with the lifestyle-modification only [45,49], by which we decided to assign the studies to the usual care comparison group. In 17 RCTs the participants were on antihypertensive medication, in six studies the participants did not take any BP lowering drugs [38,45,53–55,58]. Seven studies did not provide any information about the antihypertensive medication status of their participants [34,37,40,41,44,49,50]. Two RCTs did not report DBP values [42,57]. Seventeen studies provided HR data. Although due to false table labeling we could not use any DBP and HR data of one study [59] (Table 1).

All extracted data are presented in Table 1, which only includes studies that were found eligible, thus provides transparency about missing data. All data were extracted by two authors, Christoph Geiger and Wiebke Kathin Kohl-Heckl, between April 5th, 2024 and May 1st, 2024. No further sources but the original articles and their supporting information were used to extract the data needed for the systematic review and meta-analysis

## Blood pressure measurement

Four included studies performed 24-hour ambulatory blood pressure monitoring (24h ABPM) [54–56,59,62], whereas all others collected clinical/office BPs.

## Analyses of overall effects

Very low quality of evidence was found for positive effects of yoga on systolic blood pressure (SBP) (26 RCTs, n = 2007; MD = -7.95 mmHg, 95% CI = -10.24 to -5.66, p < 0.01; heterogeneity: I2 = 90%, $\chi^2$ = 253.38, $\tau^2$ = 26.74, P < 0.01), diastolic blood pressure (DBP) (23 RCTs, n = 1836; MD = -4.93 mmHg, 95% CI = -6.25 to -3.60, p < 0.01; heterogeneity: I2 = 92%, $\chi^2$ = 273.31, $\tau^2$ = 7.29, P < 0.01) and heart rate (HR) (14 RCTs, n = 1118; MD = -4.43 mmHg, 95% CI = -7.36 to -1.50, p < 0.01; heterogeneity: I2 = 88%, $\chi^2$ = 85.69, $\tau^2$ = 16.82, P < 0.01) compared to waitlist control (Figs 2-4).

Compared to active control, very low quality of evidence was found for positive effects yoga on SBP (5 RCTs, n = 306; MD = -4.16 mmHg, 95%CI = -10.76 to 2.44, p = 0.22; heterogeneity: I2 = 87%, $\chi^2$ = 31.88, $\tau^2$ = 47.55, P < 0.01), DBP (5 RCTs, n = 306; MD = -1.88 mmHg, 95%CI = -3.41 to -0.36, p = 0.02; heterogeneity: I2 = 0%, $\chi^2$ = 1.96, $\tau^2$ = 0.00, P = 0.74) and HR (2 RCT, n = 128; MD = -5.16 mmHg, 95% CI = -8.39 to -1.92, p < 0.01; heterogeneity: I2 = 0%, $\chi^2$ = 0.01, $\tau^2$ = 0.00, P = 0.94) (Figs 2-4).

**Table 1. Study characteristics of eligible studies.**

| Reference, Year | Origin | hypertension grade; sample size (IG/CG); weighted mean SBP/DBP | female %; caucasians% Mean age | Blood pressure assesement | Duration (intervention/ last follow-up) | Treatment group: yoga intervention description | Control group (intervention description) | Medication |
|---|---|---|---|---|---|---|---|---|
| Anjana et al., 2022 [46] | India | • Hypertension under medication and diet, 130–139/80–89 mmHg<br>• n = 65 (34/31)<br>• 135.4/85 mmHg | • 56%<br>• n/a<br>• 44.4y | office BP (BP monitor, Omron Inc. Japan) | • Intervention: 2 months<br>• Follow-up: none | • combination of OM chanting & Yoga nidra<br>• 5 days/week | • Usual care (antihypertensive diet and medication) | allowed, but no further information |
| Ankolekar et al., 2019 [34] | India | • Prehypertension<br>• n = 102 (51/51)<br>• 133.7/86 mmHg | • n/a<br>• n/a<br>• n/a | office BP (Sphygmomanometer) | • Intervention: 15 days<br>• Follow-Up: 6 months | • Asanas, Pranayamas; Meditation<br>• 15 days, 1h/d; 45min/ session; 6 days/ week. | • n/a | n/a |
| Cohen, D. L. et al., 2011 [54] | USA | • untreated SBP ≥ 130 mmHg/ < 160 mmHg, DBP < 100 mmHg<br>• n = 78 (46/32)<br>• 140/86.6 mmHg | • 50%<br>• 47%<br>• 48.2y | 24h-ABPM | • Intervention: 12 weeks<br>• Follow-up: none | • Asanas and Pranayamas<br>• 12-week Iyengar yoga (IY) program, 2 x/ week, 70min (first 6 weeks, 1 x/ week following 6 weeks) | • Enhanced Usual Care (EUC): patient-education program regarding lifestyle modification: 1h group classes meetings in weeks 1, 2, 3 and 8; 30min individual phone contact at weeks 5, 10 | not allowed |
| Cohen, D. L. et al., 2016 [55] | USA | • prehypertension to stage 1 hypertension<br>• n = 137 (43/48/46)<br>• 133.2/81.33 mmHg | • 51,1%<br>• 46,7%<br>• 47.4y | 24h-ABPM | • Intervention: 12 weeks<br>• Follow-up: 24 weeks from baseline | • Yoga intervention: 12 weeks, 2x/week, semi-private yoga classes of 90min; self-practice; 12 weeks, 90-minute community classes; Pranayama, Warm up Poses; Abdominals; Hot Poses; Cool/ Warm Down-, Deep relaxation (Shavasana) | • Blood Pressure Education Program (BPEP): health education classes (walking program 6 days/ week, 180 min/ week or 10,000 steps per day) with pedometer<br>• Combination Yoga ± BPEP (COMBO): two yoga classes per week; biweekly nutrition lecture and walking program; home practices and motivational sessions | not allowed |
| Cramer, H. et al., 2018 [62] | Germany | • essential arterial hypertension<br>• n = 75 (25/25/25)<br>• 133.5/81.6 mmHg | • 72%<br>• n/a<br>• 58.7 ± 9.5y | 24h-ABPM | • Intervention: 12 weeks<br>• Follow-Up: 28 weeks from baseline | • Yoga A: 90-min sessions 1x/week for 12 weeks; 45 min yoga postures and 45min breathing, meditation; relaxation techniques; short presentations and question-and-answer rounds<br>• Yoga B: 90-min session 1x/ week for 12 weeks; breathing, meditation; relaxation techniques, short presentations, and question-and-answer rounds (eSupplement) | • waiting list for 28 weeks - asked not to begin any yoga exercises or other new physical activities during this time<br>• at the end of the study offered to participate in an intervention corresponding to either of the other two study groups | only medicated participants included |

*(Continued)*

| Reference, Year | Origin | hypertension grade; sample size (IG/CG); weighted mean SBP/DBP | female %; caucasians% Mean age | Blood pressure assesement | Duration (intervention/last follow-up) | Treatment group: yoga intervention description | Control group (intervention description) | Medication |
|---|---|---|---|---|---|---|---|---|
| **Dhungana, R. R., et al., 2021 [51]** | Nepal | • Stage I hypertensive patients; medicated patients (SBP ≥ 130 mmHg and < 160 mmHg or DBP of ≥ 85 mmHg and < 100 mmHg) n = 121 (61/60) • 139.3/89.9 mmHg | • 47.9% • n/a • 47.7y | Office BP (sphygmomanometer (BP AG1–20, Microlife Corp., Taiwan) | • Intervention: 1 week • follow up: 90 days from baseline | • - 5-day training: postures, breathing exercise and meditation/relaxation activity structured (30 min e) | • wait list | Medication allowed; changes had to be reported; no further information |
| **Guamàn et al., 2022 [59]** | Ecuador/ Chile | • Arterial hypertension (> 140/90 mmHg), under antihypertensive medication • n = 75 (25/25/25) • 133.3/n/a mmHg | • 72% • n/a • 58.7y | 24h-ABPM | • Intervention: 12 weeks • Follow up: 28 weeks | • 2 groups, both 90min session/week by instructor, recommendation for daily home practice • Group 1: session included 45min Yoga Postures and 45 min breathing, relaxation, meditation • Group 2: session included only breathing, relacxation and meditation | • wait list | only medicated participants included; no changes monitored |
| **Hagins, M. et al., 2014 [56]** | USA | • prehypertension or stage I hypertension • n = 84 (45/39) • 134.7/80.5 mmHg | • 85,3% • n/a • 54.5y | 24h-ABPM | • Intervention: 12 weeks • Follow-up: none | • two 55-minute classes per week for 12 weeks, 3 sessions of home practice for 20 min each week • Meditation, Asana, regulated breathing, relaxation, Shavasana | • Nonaerobic Exercise: two 55-min classes per week for 12 weeks; 3 sessions of home practice for 20 minutes each week; 1. Warm-up; 2. Exercises | medicated participants; no change in medication allowed |
| Khadka, R. et al., 2010 [64] | Nepal | • Medicated hypertensives under salt-reduction • n = 14 (7/7) • 152.7/102.1 mmHg | • n/a • n/a • n/a | Office-BP | • Intervention: 6 weeks • Follow-up: none | • 6 sessions/week for 6 weeks, 0.5h/session • Strengthening exercise, Asanas, meditation, pranayama | • Usual care | All under medication (betablockers, Ca-channel-blockers) |
| **Latha et al., 1991 [35]** | India | • essential hypertensive patients • n = 14 (7/7) • 154.1/102.5 mm Hg | • n/a • n/a • n/a (45 to 70y) | Office BP (sphygnomonometer) | • Intervention: 6 months • Follow-Up: none | • 17 therapy sessions, 2x/ week for 6 months: Yoga: Breathing with arm movemen; thermal feedback added in 2nd phase of the treatment | • 1x/week: clinical setting; record BP record and general conversations • received same amount of attention as the experimental group subjects | allowed, but no further information |
| **McCaffrey, R. et al., 2014 [53]** | Thailand | • Diagnosis of hypertension (BP > 140/90 mmHg) • n = 61 (n/a) • 160.5/98.4 mmHg | • 64,8% • n/a • 56.45y | Office BP (Sphygmomanometer) | • Intervention: 8 weeks • Follow-Up: none | • 8-week pranayamas and asanas; 3x/week, approx. 63 min/session • yoga training cassettes and demonstrations • included health information (booklets) and group support in learning yogic principles and stress reduction techniques | • routine outpatient care by hospital healthcare personnel, including general education about hypertension, diet and exercise; no yoga sessions/instruction/booklet or tape | not allowed |

*(Continued)*

| Reference, Year | Origin | hypertension grade; sample size (IG/CG); weighted mean SBP/DBP | female %; caucasians% Mean age | Blood pressure assesement | Duration (intervention/last follow-up) | Treatment group: yoga intervention description | Control group (intervention description) | Medication |
|---|---|---|---|---|---|---|---|---|
| Misra, S. et al., 2019 [57] | USA | • Uncontrolled hypertension (BP ≥ 140/ ≥ 90 mmHg; ≥ 60 years older: BP ≥ 150/ ≥ 90 mmHg; diagnosed with diabetes: BP ≥ 140/ ≥ 90 mm Hg; and had BP measurements of ≤ 180/ ≤ 110mm Hg, no hypertensive urgencies) • n = 133 (101/32) • 152.5/n/a mmHg | • 48% • 89% • 60.8 ± 11.5y | Office BP | • Intervention: 6 weeks • Follow-Up: 3 months | • min. 5x/week yogic breathing exercises approx. 15min, once in a classroom setting located at the clinic, led by an instructor, min. 4x/week at home, totaling at least 5x/ week, included bellow breathing, rapid exhalations (5 min or 20 cycles), alternate nostril breathing (5 min or 20 cycles), bumblebee breathing (repeat 3 times), Om singing (repeat 3 times) | • DVD group: DVD/ YouTube group viewed video at home; performed yogic breathing exercises at least 5x/week; breathing instructions and 15-minute guided practice • Control group: record their dinner time at least 5 times a week -no other type of intervention | allowed: 6% of participants: change in BP medications; new BP medication within three months of the study start date; 80% had been on current medication for one year |
| Mourya, M. et al., 2009 [36] | India | • essential hypertension, stage 1 • n = 60 (20/20/20) • baseline BP not reported | • 48,3% • n/a • n/a (20-60y) | Office BP (mercury sphygmomanometer) | • Intervention: 3 months • Follow-Up: none | • generally, 2 weeks daily instructions in breathing exercise technique; 3 months practice, 2x/d a 15min, 10–12 hours apart; optional use of recorded cassettes or help from others to time breathing rate; yogic padmasana position to relax; breathing exercise • Group 2: Slow-breathing exercise technique, • Group 3: Fast-breathing exercise technique | • Group 1: No intervention: Patients were instructed to continue with their routine lifestyle and diet without further modification and not to practice any new yogic technique or exercise other than that prescribed during the study period. | allowed, no further information |
| Murugesan, R. et al., 2000 [37] | India | • hypertensives, no further information • n = 33 (n/a) • 156.8/108.1 mm Hg | • n/a • n/a • n/a | Office BP (standard sphygmomanometer) | • Intervention: 11 weeks • Follow-Up: none | • Experimental group I: yogic practices -morning and evening 1 h/d, 6 days/week for 11 weeks; asanas, Om recitation and meditation | • Experimental group 2: antihypertensive medication • Control group: no intervention | BP medication used in group 2; no further information |
| Pandey et al. 2023 [58] | Canada | • Hypertensive patients, no further information • n = 60 (30/30) • 128/76.5mmHg | • 30% • n/a • 63y | Office BP (Omron Hem 907XL IntelliSense Professional) | • Intervention: 3 months • Follow-up: none | • Yoga routine (two times/ week in facility, asked to do it three times/week at home) • additional 30min moderate aerobic exercise 5d/ week | • stretching routine (two times/week in facility, asked to do it three times/week at home) • Additional 30min moderate aerobic exercise 5d/week | not allowed |
| Patil, S. G. et al., 2014 [38] | India | • Grade-I hypertension • n = 60 (30/30) • 145.9/74.9 mmHg | • n/a. • n/a. • 68,9y | Office BP (mercury sphygmomanometer) | • Intervention: 3 months • Follow-Up: - | • yoga practice under supervision • 6 days/week, 1h/d in the morning for three months • Opening prayer; breathing practices, asanas or maintaining postures, pranayama or breathing exercises, cyclic meditation, devotional session, closing prayer | • -flexibility or stretching practices for 15–20 min followed by walking for 35–40 min and rest for 5 min, 6d/week, 1 h/d in the morning for three months, under supervision of an authorized instructor | not allowed |

*(Continued)*

| Reference, Year | Origin | hypertension grade; sample size (IG/CG); weighted mean SBP/DBP | female %; caucasians% Mean age | Blood pressure assesement | Duration (intervention/ last follow-up) | Treatment group: yoga intervention description | Control group (intervention description) | Medication |
|---|---|---|---|---|---|---|---|---|
| **Prakash, S. et al. 2015 [65]** | India | • Hypertensive patients<br>• n = 50 (25/25)<br>• 149.5/91 mmHg | • n/a<br>• n/a<br>• n/a | Unknown | • Intervention: 2 months<br>• Follow-up: - | • two months, 30min/d daily, Asanas | • Usual care | Allowed, both groups used it |
| **Pushpanatan, P. et al., 2016 [39]** | India | • hypertensive patients, no further information<br>• n = 80 (40/40)<br>• 125.3/81.8 mmHg | • 20%<br>• n/a.<br>• 43,4y | Office BP | • Intervention: 12 weeks<br>• Follow-Up: - | • 12 weeks of yoga therapy, 3x45min weekly by a trained yoga teacher<br>• Patients were motivated to practice the same daily at home<br>• brief prayer, preparatory practices, asanas, pranayama, shavasana<br>• Some classes received additionally talks on diet/lifestyle modification in controlling chronic lifestyle disorders | • allopathic medicines only | allowed, both groups used it |
| **Saptharishi, L. G. et al., 2009 [40]** | India | • hypertensives and prehypertensives,<br>• n = 120 (n/a)<br>• 125.5/84.6 mmHg | • 33,33%;<br>• n/a.;<br>• 22.5 ± 1.3y | Office BP (mercury sphygmomanometer) | • Intervention: 8 weeks<br>• Follow-Up: - | • Group IV (Yoga): yoga lessons by a qualified yoga teacher<br>• encouraged to practice yoga for 30–45 minutes per day for at least five days a week.<br>• asanas, relaxation techniques (pranayama (= breathing exercise) | • Group I (Control): No intervention<br>• Group II (Physical exercise): brisk walking for 50–60 min/d, 4d/ week<br>• Group III (Salt Intake reduction): reduce daily salt intake to at least half of their previous intake. | n/a |
| **Shantakumari, N. et al., 2012 [41]** | India | • (pre)hypertensive patients with known type 2 diabetes<br>• n = 100 (50/50)<br>• 139.5/89.4 mmHg | • 48%;<br>• n/a.;<br>• 45y | Office BP (mercury sphygmomanometer) | • Intervention:2 weeks<br>• Follow-Up: 3 months | • Experimental group: structured yogic exercises 1h daily for two weeks instructed by experienced yoga teacher<br>• encouraged to practice yoga at home<br>• sulfonylureas for a period of 3 months | • Control group: sulfonylureas for a period of 3 months | n/a |
| Shetty, P. et al., 2017 [42] | India | • hypertensives and prehypertensives<br>• n = 60 (30/30)<br>• 150.9/n/a mmHg | • n/a<br>• n/a<br>• n/a (25–65 years) | Office BP (mercury sphygmomanometer) | • Intervention: 30 days<br>• Follow-Up: - | • Pranayama: 30 days max., Sheetali and Sheetkari pranayamas; Each pranayama was demonstrated and practiced for 10 minutes each day and observation by members of the research team | • wait list control: were asked to sit quietly for 20 minutes daily, attendance record on a daily basis | all participants on BP medication, changes not monitored |
| Shetty S. et al., 2022 [48] | India | • Arterial Hypertension (>130/80 mmHg)<br>• n = 65 (33/32)<br>• 142.6/92 mmHg | • 48,2%<br>• n/a<br>• 49,4y | Office BP (mercury sphygomanometer) | • Intervention: 3 months<br>• Follow-up: - | 10 days intensive training (1h) followed by practice continuance for 3 months, 6 days/week<br>• prayers, asanas, pranayama | • wait list | All participants on BP medication, changes not monitored |

*(Continued)*

**Table 1.** (Continued)

| Reference, Year | Origin | hypertension grade; sample size (IG/CG); weighted mean SBP/DBP | female %; caucasians% Mean age | Blood pressure assesement | Duration (intervention/ last follow-up) | Treatment group: yoga intervention description | Control group (intervention description) | Medication |
|---|---|---|---|---|---|---|---|---|
| Singh V. et al 2022 [49] | India | • Prehypertension<br>• n = 238 (117/121)<br>• 127.8/83 mmHg | • 50,4%<br>• n/a<br>• 51,2y | n/a | • Intervention: 2 months<br>• Follow-up: 3 and 6 months | • Life-Style-Modification (LSM) for BP reduction ± Yoga: 4 supervised Yoga Classes for 2 months and asked to practice at home<br>• Yoga booklets<br>• Asanas and Pranayama | • Life-style Modification (LSM) for BP reduction | patients excluded with drug modifications within 2 months pre intervention |
| Sujatha, T., Judie, A., 2014[43] | India | • Stage 1 and Stage 2 hypertension and those receiving BP medication<br>• n = 238 (118/120)<br>• 152.8/94.6 mmHg | • 53,8%<br>• n/a<br>• n/a (30–60 y) | Office BP (mercury sphygmomanometer), 2 readings | • Intervention: 12 weeks<br>• Follow-Up: - | • 5-day intensive continuous training for 2 h/day, then daily yoga practice for 30–45 mins at home, at least for 5d/week<br>• instruction through DVD, attend the yoga group session once in two weeks<br>• asanas, pranayama, meditational practices | • usual care | all participants on BP medication, change not monitored |
| Thanalakshmi, J. et al., 2020 [44] | India | • primary hypertension<br>• n = 100 (50/50)<br>• 145/88 mmHg | • 28,75%<br>• n/a<br>• 38.5y | Office BP (Omron Hem 7130L), 3 recordings | • Intervention: 3 months<br>• Follow-Up: - | • instruction and supervision of pranayama intervention by qualified yoga and naturopathy doctor<br>• practice pranayama for 30 min/d for 4 weeks between 7:00–9:00 a.m. | • no treatment | n/a |
| Thiyagarajan, R. et al., 2015 [45] | India | • prehypertensives<br>• n = 192 (96/96)<br>• 127/85 mmHg | • 61%<br>• n/a<br>• 43.3y | Office BP, (automatic BP monitor CH432B, Citizen Systems Japan, Tokyo), 3 readings | • Intervention: 12 weeks<br>• Follow-Up: - | • Life-style-modification (LSM) ± Yoga: Yoga therapy classes by qualified yoga teachers, sessions of 45 mins, 3x/week under supervision, for 12 weeks; home practice on remaining days of the week<br>• warm up, asanas, pranayamas, meditation/relaxation | • LSM group (active control): reduce: dietary salt intake, alcohol consumption; increase fruit and vegetable content in diet as per DASH (dietary approach to stop hypertension) criteria, aerobic physical activity (<br>• 30 min/d, reduce/maintain body weight, diary of their daily dietary intake and physical activity | not allowed |
| Tolbaños Roche, L., Mas Hesse, B., 2014[60] | Spain | • essential arterial hypertension<br>• n = 50 (25/25)<br>• 140.5/82.4 mmHg | • 70%<br>• n/a<br>• 57.5 ± 7.82y | Office BP (Omron sphygmomanometers) | • Intervention: 13 weeks<br>• Follow-Up: - | • yoga practice program 2 d/week for 3 months (=26 90-min sessions)<br>• asanas; breathing techniques specifically indicated in arterial hypertension; relaxation; visualization practices; meditation; mindfulness exercises | • no treatment | medication allowed, but no changes in medication during intervention period (exclusion) |

*(Continued)*

**Table 1.** (Continued)

| Refer-ence, Year | Origin | hypertension grade; sample size (IG/CG); weighted mean SBP/ DBP | female %; caucasians% Mean age | Blood pressure assesement | Duration (intervention/ last follow-up) | Treatment group: yoga intervention description | Control group (inter-vention description) | Medication |
|---|---|---|---|---|---|---|---|---|
| Tolbaños Roche, L. 2017 [61] | Spain | • essential arterial hypertension<br>• n = 85 (meditation 21/ pranayama 23/ Yoga 22/ control 19)<br>• baseline BP not reported | • 64%<br>• n/a<br>• 54,73 ± 9,2y | Office BP (Omrom sphygmo-manome-ters) | • Intervention: 8 weeks<br>• Follow-Up: - | • 2d/week for two months, total of 15 sessions.<br>• warm-up, asanas; pran-ayama or yoga breathing techniques; shavasana relaxation or body scan meditation; talks about mindfulness in daily life | • n/a | medication allowed, no changes monitored |
| Wahyuni, N. et al., 2020 [50] | Bali, Indo-nesia | • hypertensive dia-betic population<br>• n = 39 (21/18)<br>• 154.9/92,9 mmHg | • 57%<br>• n/a<br>• intervention: 63.1y | Office BP (Onemed® sphygmo-manometer) | • Interven-tion: 8 weeks<br>• Follow up: - | • 2 times training/week 60min<br>• Asanas; Shavasana | • no treatment | n/a |
| Wolff, M. et al., 2016 [63] | Swe-den | • diagnosed hyper-tension, high normal or grade 1 hypertension<br>• n = 191 (96/95)<br>• 149.4/88.2 mmHg | • 51,8%<br>• n/a<br>• 64,7y | Office BP, (Omron 705-IT, Omron Health Care Co., Kyoto, Japan), 2–3 readings | • Interven-tion: 12 weeks<br>• Follow-Up: - | • information and instruc-tions concerning two yoga exercises during single 30 min GP consul-tation, 15 min 2x/d | • treatment as usual (prior medication) | Allowed; no changes monitored |

IG = intervention group: CG = control group; ABPM = ambulatory blood pressure measurement

Ten RCTs reported safety data [44,46,48,51,54–56,59,62,63]. Among them, in eight no or non-serious adverse events occurred. Two RCTs reported adverse events but did not further specify [44,54].

## Subgroup and sensitivity analyses

Compared with waitlist control, effects were found on SBP, DBP and HR for RCTs that included hypertensive partic-ipants, RCTs including physical postures, RCTs using clinical BP measurement or RCTs originated in India. Effects on SBP and DBP but not on HR were found for RCTs that did not include physical postures, RCTs that allowed co-medication, RCTs that included prehypertensive participants or RCTs not originated in India. Effects on DBP and HR but not on SBP were found for RCTs measuring 24h ABPM. Effects on DBP but not on SBP and HR were found for RCTs including a mixed sample with both prehypertensive and hypertensive participants or RCTs that did not allow co-medication (S2 Table).

Compared with active control, an effect was found on SBP only for the RCTs including hypertensive participants. Effects on DBP and HR but not on SBP were found for RCTs with exercise as active control intervention. Effects on DBP but not on SBP and HR were found for RCTs including hypertensive participants, RCTs not allowing co-medication, RCTs measuring clinical BP. (S3 Table).

Compared with any control intervention effects were found on SBP, DBP and HR for RCTs measuring clinical BP. An effect on DBP but not on SBP or HR was found for RCTs measuring 24h ABPM. (S4 Table).

In sensitivity analyses of yoga compared with waitlist control the effects on SBP, DBP and HR did persist after excluding studies with diabetic participants [41,50,57] and one study with missing data about group sizes [53]. The effects also persisted after excluding studies with a duration of less than 8 weeks of their yoga intervention (S5 Table).

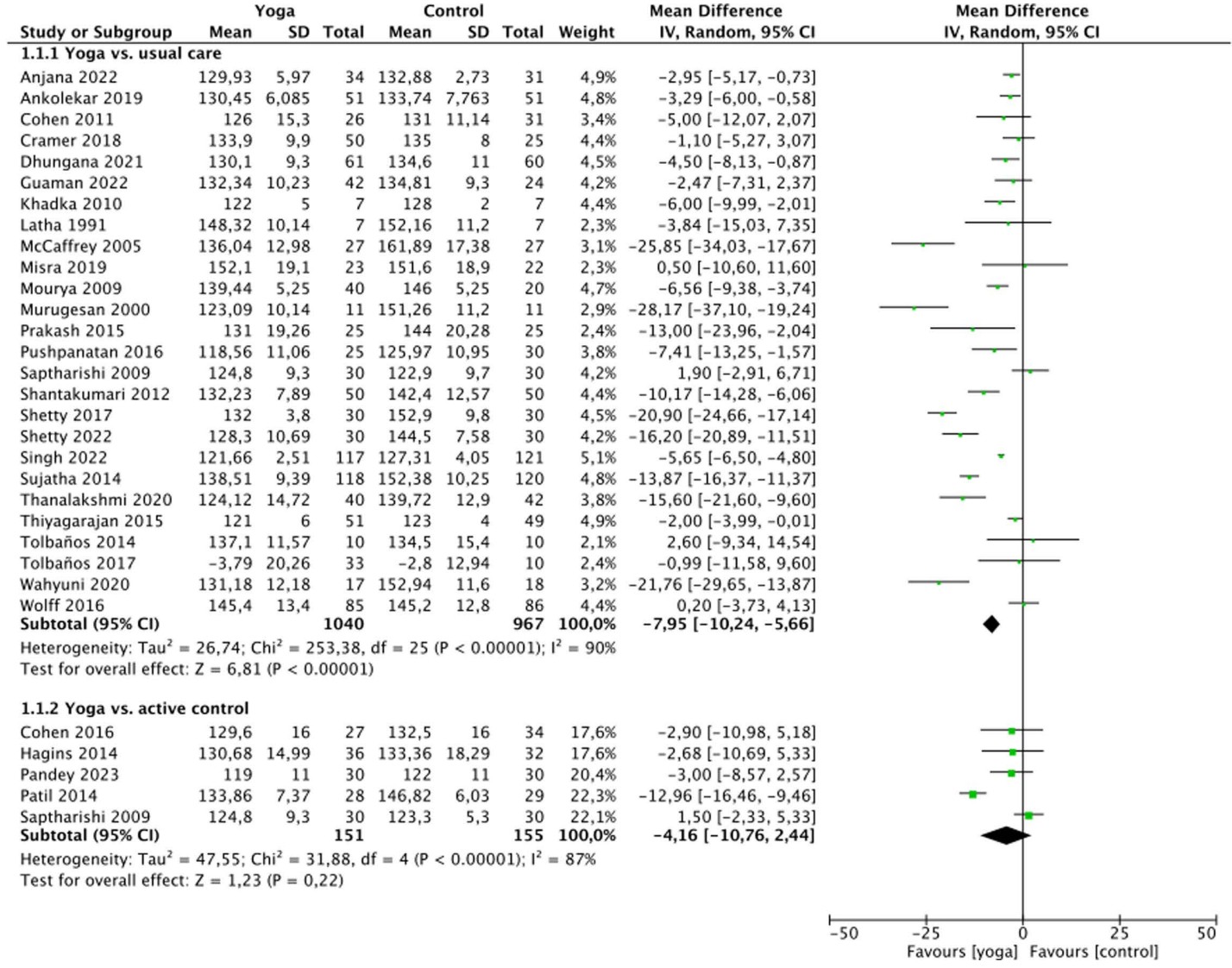

**Fig 2. Forest plot for systolic blood pressure.**

None of the subgroup analyses could reduce heterogeneity substantially. In sensitivity analyses, the effects of yoga compared with waitlist or active control on SBP, DBP and HR were robust against selection, detection, and attrition bias.

## Publication Bias

In funnel plots visual asymmetry was present for SBP, DBP and HR, suggesting publication bias. Using Egger's test, no publication bias could be identified for SBP (p = 0.80), DBP (p = 0.28) and HR (p = 0.83). (S1-S3 Figs).

## Moderator analyses

Meta-regression did not reveal a significant impact of total duration of yoga interventions on SBP (p = 0.51), DBP (p = 0.55) and HR (p = 0.08) for studies comparing yoga to usual care. (S4-S6 Figs).

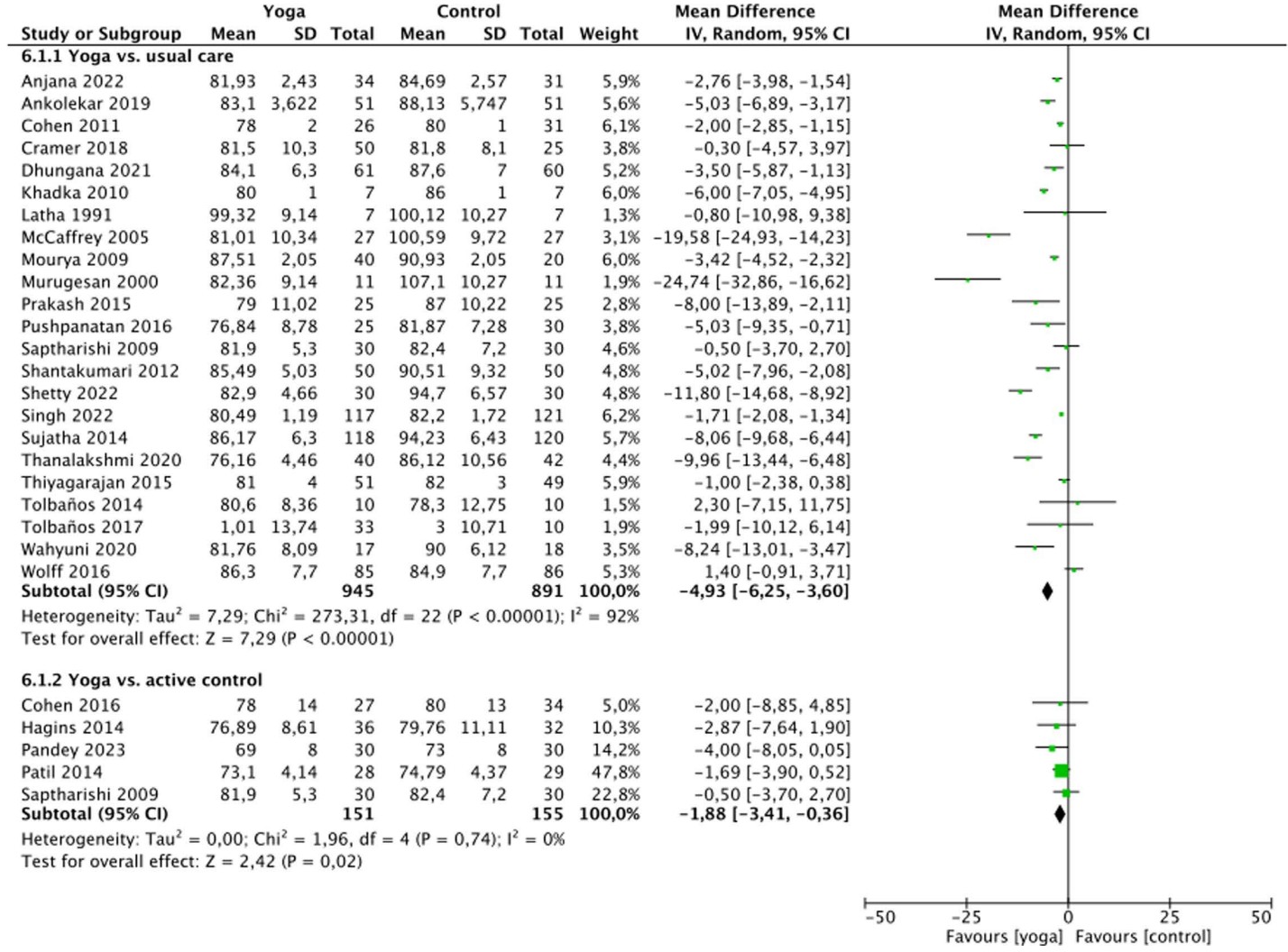

**Fig 3. Forest plot for diastolic blood pressure.**

## Discussion

This systematic review and meta-analysis aimed to identify whether yoga in general and different types of yoga interventions in particular can be beneficial in addition to standard AHT treatment. 30 RCTs among prehypertensive to hypertensive patients were included in the analysis. SBP, DBP and HR were reduced by 7.95 mmHg, 4.93 mmHg and 4.43 bpm in studies comparing yoga interventions to waitlist control/usual care. When compared to any active control, we found a reduction of SBP, DBP and HR by 4.16 mmHg, 1.88 mmHg and 5.16 bpm. In subgroup analyses, positive effects on SBP and DBP were found on either pre- or hypertensive participants when co-medication was allowed. The results were found regardless of the country in which the study was conducted (India vs. any other country) or whether the yoga intervention did or did not include physical postures, but only when clinical BP was measured. In the mixed group of prehypertensive to hypertensive participants and when 24h-ABPM was measured, positive effects were just seen for DBP. Compared to active control, studies with hypertensive participants only showed a positive effect on SBP. Although these results on BP levels and HR were stable in sensitivity analyses, considerable heterogeneity remained.

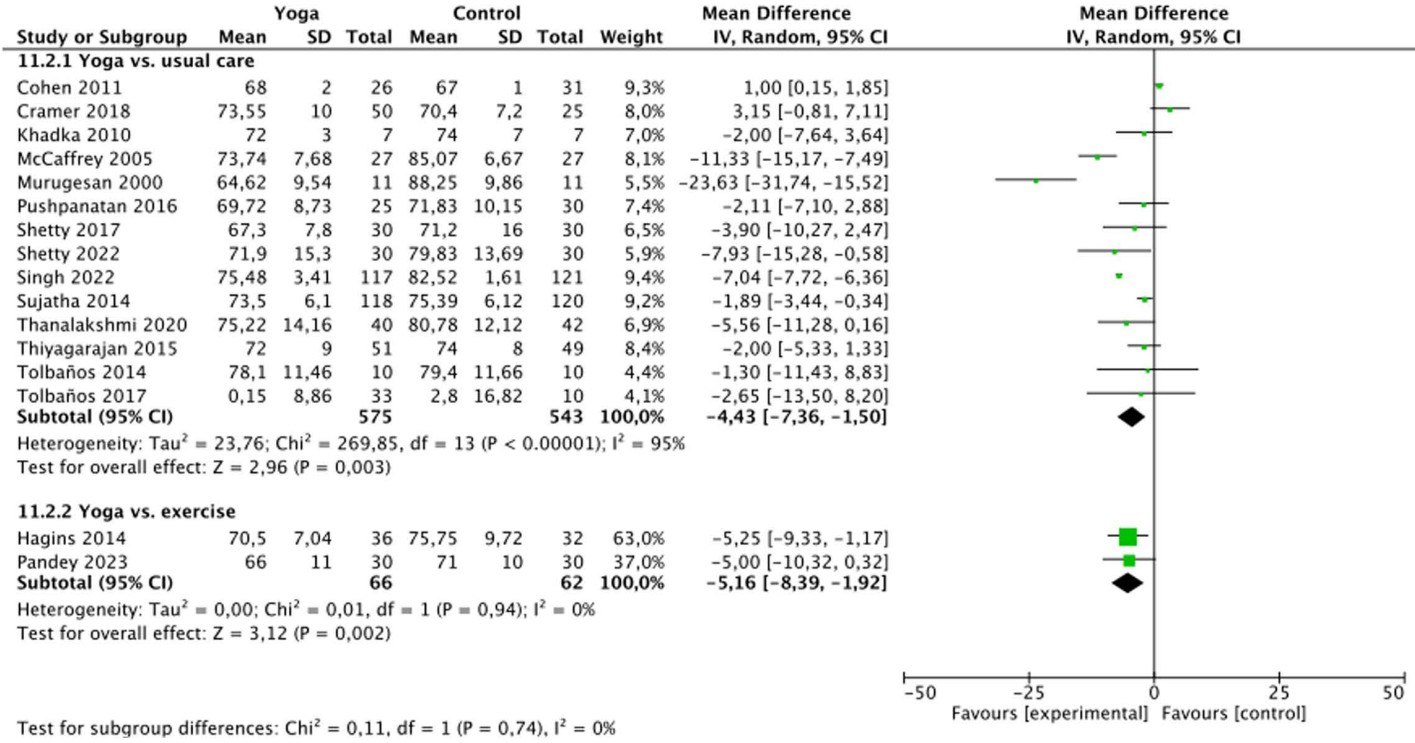

**Fig 4. Forest plot for heart rate.**

This heterogeneity of our included studies is based on multiple influencing factors all remaining on study design. As summarized in Table 1, some RCTs based their interventions on breathing exercises or physical exercises only, while others included multiple aspects of yoga. As well, studies showed various durations and only some collected follow-up data. Further, gender and age distribution varied.

This work represents an update of a previous systematic review and meta-analysis published by Cramer et al. in 2014. At that time, seven RCTs met the inclusion criteria, and emerging but low-quality evidence was found for yoga as an adjunctive therapy in AHT. These findings were reproducible throughout our update, but still only low-quality evidence was found. Even though our results show the promising potential of yoga as a concomitant treatment in AHT, limitations in study quality remain. From our point of view, a clear limitation of the study protocols is that most RCTs based their results on clinical/office BP measurements. Today, guidelines recommend at least two to three clinical BP measurements on different days within one to four weeks to confirm the diagnosis of AHT [8]. The 24h-ABPM represents another, even more reproducible tool and also helps to identify white coat hypertension or masked hypertension. Additionally, 24h-ABPM is used to monitor the therapeutic effect of antihypertensive medication or other interventions that may lead to reduction in BP [66]. Only four of the included studies performed 24-ABPM. Effects on all parameters, SBP, DBP and HR, were only found when clinical measurements were performed, whereas in 24h-ABPM an effect remained only for DBP. Regarding cardiovascular risk, SBP and its reduction seem to be the most important factor in reducing cardiovascular risk, especially over the age of 60 [67,68]. While a reduction in DBP may be beneficial in younger individuals [67], the overall effect on the majority of hypertensive individuals likely remains limited. The effect found in our meta-analysis may be due to the small number of studies performing 24h-ABPM, limiting its power. For better comparability, future studies should include 24h-ABPM.

**Table 2. Risk of bias.**

| Reference, Year | Random sequence generation (selection bias) | Allocation concelement (selection bias) | Blinding of Participants and Personnel (performance bias) | Blinding of outcome assessement (detection bias) | Incomplete outcome data (attrition bias) | Selective Reporting (reporting bias) | Other bias - exclusion criteria given | Other bias - Funding |
|---|---|---|---|---|---|---|---|---|
| Anjana et al., 2022 [46] | Low | Low | Unclear | Unclear | Low | Low | Low | Low |
| Ankolekar et al., 2019 [34] | Unclear | Unclear | Unclear | Unclear | Unclear | High | Unclear | Unclear |
| Cohen, D. L. et al., 2011 [54] | Unclear | High | High | High | High | Low | Low | Low |
| Cohen, D. L. et al., 2016 [55] | Low | High | High | High | High | Low | Low | Low |
| Cramer, H. et al., 2018 [62] | Low | Low | Unclear | Low | Low | Low | Low | Low |
| Dhungana, R. R., et al.2021 [51] | Low | Low | High | High | Low | Low | Low | Low |
| Guamàn et al., 2022 [59] | High | Unclear | Unclear | Low | Low | High | Low | Unclear |
| Hagins, M. et al., 2014 [56] | Low | Low | High | Low | Low | Low | Low | Low |
| Khadka, R. et al., 2010 [52] | High | Unclear | Unclear | Unclear | Low | Low | Low | Unclear |
| Latha et al., 1991 [35] | High | Unclear | Unclear | Low | High | High | Unclear | Unclear |
| McCaffrey, R. et al., 2014 [53] | Low | Unclear | Unclear | Unclear | Low | Low | Unclear | Unclear |
| Misra, S. et al., 2019 [57] | Low | Low | Unclear | Unclear | High | Low | Low | Low |
| Mourya, M. et al., 2009 [36] | Low | Low | High | Unclear | Low | Low | Low | Low |
| Murugesan, R. et al., 2000 [37] | Unclear | Unclear | Unclear | Unclear | Unclear | Low | Unclear | Unclear |
| Pandey et al. 2023 [58] | High | Unclear | Unclear | Unclear | Unclear | Low | Low | Low |
| Patil, S. G. et al., 2014 [38] | Low | Unclear | Unclear | Low | Low | Low | Low | Low |
| Prakash, S. et al. 2015 [47] | High | Unclear | Unclear | Unclear | Unclear | Low | Low | Low |
| Pushpanatan, P. et al., 2016 [39] | Low | Low | Unclear | Unclear | Low | Low | Low | Unclear |
| Saptharishi, L. G. et al., 2009 [40] | Low | Unclear | Unclear | Unclear | Low | Low | Unclear | Low |
| Shantakumari, N. et al., 2012 [41] | High | Unclear | Unclear | Unclear | Low | Low | Low | Unclear |
| Shetty, P. et al., 2017 [42] | Low | Low | High | Low | High | High | Low | Unclear |
| Shetty S. et al., 2022 [48] | Low | Low | High | Unclear | Low | Low | Low | Low |
| Singh V. et al 2022 [49] | Low | Unclear | Unclear | Unclear | Low | Low | Low | Low |
| Sujatha, T., 2014 [43] | Unclear | Unclear | Unclear | Unclear | Low | Low | Low | Unclear |
| Thanalakshmi, J. et al., 2020 [44] | Low | Low | High | Low | Low | Low | Low | Unclear |
| Thiyagarajan, R. et al., 2015 [45] | Low | High | High | Unclear | High | Low | Low | Low |
| Tolbaños Roche, L. 2014 [60] | Unclear | Unclear | Unclear | Unclear | High | Low | Unclear | Low |
| Tolbaños Roche, L. 2017 [61] | High | Unclear | Unclear | Unclear | High | Low | Unclear | Low |
| Wahyuni, N. et al., 2020 [50] | Low | Low | Unclear | Unclear | Low | Low | Low | Low |
| Wolff, M. et al., 2016 [63] | Low | Low | Unclear | Low | Low | Low | Low | Low |

Another publication by Zhu et al., 2022, also documented positive effects on AHT for various physical exercises – including yoga interventions. With the exception of one RCT, that used not only yoga but an entire mindfulness program, our research was based on the same and several other RCTs [69].

In 2019, Wu et al. [22] published a systematic review and meta-analysis on yoga and hypertension. In the 49 included studies, yoga was shown to have a positive effect on both SBP (-5,0 mmHg) and DBP (-3,9 mmHg), although in sub-group analyses on hypertensives and prehypertensives the effects varied (SBP/DBP approximately 9/5 mmHg and 5/3 mmHg, respectively). Overall, only studies in which the control intervention was based on a non-exercise, non-diet group and BP measurements were reported before and after the intervention were included in their analysis. Not all included studies performed their intervention in prehypertensive or hypertensive participants. Hypertensive and non-hypertensive individuals may show different responses in BP lowering, as a reduction of BP in healthy participants does not appear

to be necessary from a therapeutical perspective. Including healthy as well as hypertensive individuals in such an analysis might therefore mask potential effects of yoga interventions on hypertensives. In addition, it seems interesting to determine whether yoga can outperform other forms of physical activation or other active programs that lead to lifestyle change. Our results show an effect compared to active control but further investigation is needed because of the small number of participants included.

While our data show an effect for yoga on BP in both prehypertensive and hypertensive groups considered separately, there were no significant changes in SBP in mixed groups, only for DBP. This circumstance is probably due to the small number of studies available as well, as only two studies included a mixed population of prehypertensive and hypertensive individuals. As mentioned earlier, in prehypertensive individuals only promising effects for SBP and DBP reduction were found. The sample in this subgroup analysis, and therefore the results, too, is identical to the studies Khandekar et al. used in their analysis in 2021 [70]. Hence, we suggest that yoga could already be used in primary prevention in a population at high risk for developing manifest AHT and associated cardiovascular complications. Assuming that yoga is beneficial, the literature still seems to lack the right combination of physical postures and breathing techniques. The studies we included were heterogeneous in terms of yoga application and we do not know to what proportion the effects on BP were due to the implementation of breathing techniques/meditation or physical postures. Nor can we say whether any type of yoga in a group of prehypertensive participants provides more health benefit to the individual than any other type of exercise. It is also interesting to note that the studies were conducted in many different countries. As study origin is known to potentially impact results as previously described by Cramer et al. [71], we compared the effects of yoga in studies from India vs. any other country but found them to be independent of the study origin.

We hypothesized that yoga interventions of longer duration would have larger effects on SBP, DBP and HR, unfortunately our meta-regression did not support our assumption. There was no significant correlation between total duration of yoga interventions and lowering effects on SBP, DBP and HR for studies comparing yoga to usual care. Publication bias only was present for the outcome HR, which can partly be explained by the small sample of just eleven studies for this outcome.

In conclusion, yoga could be a useful adjunct in the treatment of AHT. We found promising results even when participants were on medication. In prehypertensive individuals, we also see potential for lowering BP levels.

Our findings highlight a statement of the International Society of Hypertension (ISH) that analyzed various mind-body medicine approaches as additional treatment options in individuals with AHT. Among other approaches, yoga was highlighted as beneficial for lowering BP levels [8].

The mechanisms affecting the BP are various. On the one hand it has stress-reducing effects that are most likely due to a regulation of the function of the autonomic nervous system [13]. Besides, the physical postures, depending on the intensity and duration of hold, may result in an optimized cardiac function and muscular function, if executed regularly. This may lead to an overall improved physical function, facilitating daily life.

Our results suggest that yoga may be a potential adjunct for both therapy and primary prevention of AHT, but so far, many questions remain unanswered. Because of the small number of studies, we still need more knowledge about the dosage, the combination of yoga elements, and the target population for whom yoga may be most beneficial. As well, it is important to evaluate which population may be interested in incorporating yoga into a daily lifestyle for lowering blood pressure. This could for example be achieved by qualitative studies such as patient-focused interviews.

Most studies based their results on relatively healthy individuals who had no other disease besides AHT. Autonomous nervous system reactivity could be different compared to individuals with underlying health problems.

Even though we only see low-quality evidence so far based on methodological aspects mentioned in the limitations´ section, yoga may play an important role in controlling BP. Yoga has the advantage to meet multiple mechanisms for lowering BP, depending on its incorporated aspects such as relaxation, meditation, breathing and physical activation.

Especially in prehypertensive individuals, where first-line treatment should be based on lifestyle-modification before including medication, yoga can play an important role. It meets these various aspects of optimizing the autonomous nervous system and includes physical activation, all with the potential of lowering BP. In hypertensive individuals, it can be added as a low-cost therapeutic approach, that can also be performed from home after short period of training or supported by standardized video/online courses, if available.

## Limitations

Despite subgroup and sensitivity analyses considerable heterogeneity within and between included studies was present in several domains. High and unclear risk of bias was present in some studies in different categories (Table 2). Therefore, our results should be interpreted with caution. General limitations are based on methodological reasons, incomplete reporting and a low quality of study performance in general. These limitations include different BP measurement techniques, a wide range of BP levels, variation in (control) interventions, e.g., passive vs. active control, type of yoga, duration of the intervention period, minutes of yoga applied, as well as medication status, characteristics of study populations and more. To be able to further analyze the role of yoga in controlling BP, future studies need to meet comparable criteria, such as 24h-ABPM, more standardized programs and durations. As well, it would be helpful to integrate long follow-up periods to evaluate long-term effects and duration of such.

Although the number of studies available has been growing throughout the last decade, the total number of studies on this topic and the number of participants included remains small and sensitivity analysis is limited. Even though the safety of yoga is likely high, only seven RCTs provided safety data.

## Conclusion

In our systematic review we aimed to evaluate the effects of yoga on lowering blood pressure in a population of prehypertensive and hypertensive participants. Our meta-analysis suggests that yoga can lower elevated blood pressure, although very low quality of evidence was found for lowering SBP, DBP and HR for both waitlist and active control interventions. Considerable differences in medication status, BP measurement techniques could potentially influence the outcomes.

In conclusion, yoga may be useful therapeutic means in the management of arterial hypertension and prehypertension and should be integrated into a multimodal treatment approach. Although the number of studies available has been growing throughout the last decade, the number of included studies remains small and heterogeneity between studies is high. More and larger high-quality studies are required to substantiate our findings. Studies should preferably include 24h-ABPM to enable effective and replicable BP evaluation. Also, a closer look at the effects in participants treated with antihypertensive medication could be worthwhile to develop future integrative approaches.

## Supporting information

**S1 Table. Excluded studies.**
(XLSX)

**S2 Table. Subgroup analyses of yoga vs. waitlist control.**
(DOCX)

**S3 Table. Subgroup analyses of yoga vs. active control.**
(DOCX)

**S4 Table. Subgroup analyses of yoga vs. any control.**
(DOCX)

**S5 Table. Sensitivity analyses of yoga vs waitlist control.**
(DOCX)

**S6 Table. Risk of Bias Evaluation.**
(DOCX)

**S7 Table. Quality of Evidence Assessment.**
(DOCX)

**S8 Table. Results Quality of Evidence Assessment.**
(DOCX)

**S9 Table. Prisma Checklist.** Linear regression test of funnel plot asymmetry.
(PDF)

**S1 Fig. Funnel plot of yoga vs usual care for SBP.**
(TIF)

**S2 Fig. Funnel plot of yoga vs usual care for DBP.**
(TIF)

**S3 Fig. Funnel plot of yoga vs usual care for HR.**
(TIF)

**S4 Fig. Bubble plot, univariate meta-regression of yoga vs usual care for SBP.**
(TIF)

**S5 Fig. Bubble plot, univariate meta-regression of yoga vs usual care for DBP.**
(TIF)

**S6 Fig. Bubble plot, univariate meta-regression of yoga vs usual care for HR.**
(TIF)

## Author contributions

**Conceptualization:** Christoph Geiger, Wiebke Kathrin Kohl-Heckl.

**Data curation:** Christoph Geiger, Holger Cramer, Dennis Anheyer, Gustav Dobos, Wiebke Kathrin Kohl-Heckl.

**Formal analysis:** Holger Cramer, Dennis Anheyer, Wiebke Kathrin Kohl-Heckl.

**Investigation:** Christoph Geiger, Wiebke Kathrin Kohl-Heckl.

**Methodology:** Christoph Geiger, Holger Cramer, Dennis Anheyer, Wiebke Kathrin Kohl-Heckl.

**Supervision:** Gustav Dobos.

**Validation:** Dennis Anheyer, Gustav Dobos, Wiebke Kathrin Kohl-Heckl.

**Visualization:** Christoph Geiger.

**Writing – original draft:** Christoph Geiger, Wiebke Kathrin Kohl-Heckl.

**Writing – review & editing:** Christoph Geiger, Holger Cramer, Gustav Dobos, Wiebke Kathrin Kohl-Heckl.

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
