## [Decision Letter · Decision Letter 0]

17 Oct 2024

PONE-D-24-24473A Systematic Review and Meta-Analysis of Yoga for Arterial HypertensionPLOS ONE

Dear Dr. Geiger,

Thank you for submitting your manuscript to PLOS ONE. After careful consideration, we feel that it has merit but does not fully meet PLOS ONE’s publication criteria as it currently stands. Therefore, we invite you to submit a revised version of the manuscript that addresses the points raised during the review process.

We look forward to receiving your revised manuscript.

Kind regards,

Bhabesh Deka

Academic Editor

PLOS ONE

“Wiebke Kohl-Heckl is a member of a post-doc scholarship by the Karl und Veronica Carstens Foundation Germany. The work itself did not receive any specific funding”

4. We notice that your supplementary tables are included in the manuscript file. Please remove them and upload them with the file type 'Supporting Information'. Please ensure that each Supporting Information file has a legend listed in the manuscript after the references list.

5. Please remove your figures from within your manuscript file, leaving only the individual TIFF/EPS image files, uploaded separately. These will be automatically included in the reviewers’ PDF.

6. We notice that your supplementary figures are uploaded with the file type 'Figure'. Please amend the file type to 'Supporting Information'. Please ensure that each Supporting Information file has a legend listed in the manuscript after the references list.

7. As required by our policy on Data Availability, please ensure your manuscript or supplementary information includes the following:

Additional Editor Comments:

Authors must address the second reviewers concerns carefully one by one before the manuscript can be further considered for reviewing.

Reviewers' comments:

Reviewer's Responses to Questions

**Comments to the Author**

1. Is the manuscript technically sound, and do the data support the conclusions?

Reviewer #1: Yes

Reviewer #2: Yes

2. Has the statistical analysis been performed appropriately and rigorously? 

Reviewer #1: Yes

Reviewer #2: Yes

3. Have the authors made all data underlying the findings in their manuscript fully available?

Reviewer #1: Yes

Reviewer #2: Yes

4. Is the manuscript presented in an intelligible fashion and written in standard English?

Reviewer #1: Yes

Reviewer #2: Yes

5. Review Comments to the Author

Reviewer #1: A Systematic Review and Meta Analysis of Yoga for Arterial Hypertension

I have studied the systematic Review and Meta Analysis presented to me in detail and found it to be quite interesting and in my personal opinion a valuable piece of work. I highly appreciate the work of the researchers and would like to congratulate them in this regard. It will form the basis of future studies.

The Study found a significant effect on various variable of Arterial Hypertension and in the subgroup analysis it found the following details :-

Subgroup Analysis

The study found that as compared with waitlist control, effects were found on SBP, DBP and HR for RCTs that included hypertensive participants, RCTs including physical postures, RCTs using clinical BP measurement or RCTs originated in India.

Effects on SBP and DBP but not on HR were found for RCTs that did not include physical postures, RCTs that allowed co-medication, RCTs that included prehypertensive participants or RCTs not originated in India.

Effects on DBP and HR but not on SBP were found for RCTs measuring 24h ABPM.

Effects on DBP but not on SBP and HR were found for RCTs including a mixed sample with both prehypertensive and hypertensive participants or RCTs that did not allow co-medication

Compared with active control, an effect was found on SBP only for the RCTs including hypertensive participants. Effects on DBP and HR but not on SBP were found for RCTs with exercise as active control intervention. Effects on DBP but not on SBP and HR were found for RCTs including hypertensive participants, RCTs not allowing co-medication, RCTs measuring clinical BP and.

Compared with any control intervention effects were found on SBP, DBP and HR for RCTs measuring clinical BP. An effect on DBP but not on SBP or HR was found for RCTs measuring 24h ABPM.

My observation is that this study can form a base for future studies. However, future studies should work on the specific aspects of yoga practises and some standardization needs to be evolved. This can help in generalization and also help in evolution of a yoga based clinical model for hypertension.The specific aspects can be a yoga based life style, Asanas – Physical postures, Pranayama- Breathing practices, Dhyana- Meditation , Relaxation exercises or practises like Yog nidra etc. The repetitions, durations, and quality needs to be uniform as far as possible.The researchers have said that Yoga may be useful in the management of arterial hypertension and pre-hypertension. It will work both as cure and prevention. It can be integrated into a multi-modal approach . Future studies should use 24 hours ABPM and lot of standardization needs to be adopted. I would suggest a holistic research approach in which both Doctors and Yoga practioners, researchers, Yoga specialist need to come together for future research.

From my side I would approve the publication of the systematic review. 

Reviewer #2: Dear Author,

Overall, the authors have conducted a comprehensive and well-executed systematic review and meta-analysis. The study's strengths include a thorough search strategy, clear inclusion/exclusion criteria, and a robust assessment of risk of bias. The findings are generally consistent with previous research and provide valuable insights into the potential benefits of yoga for managing prehypertension and hypertension.

Specific Comments:

Heterogeneity: The authors acknowledge the high degree of heterogeneity among the included studies. It would be beneficial to explore potential sources of heterogeneity, such as differences in yoga styles, duration of interventions, and participant characteristics. This could help identify subgroups that may respond differently to yoga.

Dose-Response Relationship: The authors might consider investigating a dose-response relationship between the intensity or duration of yoga interventions and blood pressure reduction. This could provide valuable information for tailoring yoga programs to individual needs and preferences.

Quality of Evidence: While the authors note the very low quality of evidence for some of their findings, it's important to discuss the implications of this for clinical practice and future research. Are the observed effects still clinically relevant despite the low quality of evidence? What steps can be taken to improve the quality of future studies?

Long-Term Effects: Given the chronic nature of hypertension, it would be valuable to explore the long-term effects of yoga on blood pressure management. This could include studies with longer follow-up periods to assess the sustainability of the observed benefits.

Cost-Effectiveness: The authors might consider discussing the cost-effectiveness of yoga as a treatment option for prehypertension and hypertension. This could help inform decision-making by healthcare providers and policymakers.

Additional Suggestions:

Subgroup analyses: Consider conducting subgroup analyses based on factors such as age, sex, and baseline blood pressure levels. This could help identify populations that may benefit most from yoga.

Mechanism of Action: Discuss potential mechanisms through which yoga may lower blood pressure, such as stress reduction, improved cardiovascular fitness, or alterations in the autonomic nervous system.

Comparison with Other Lifestyle Interventions: Compare the effects of yoga with other lifestyle interventions, such as exercise, dietary changes, and stress management techniques.

Patient Preferences: Explore patient preferences for yoga as a treatment option. This could help identify factors that may influence adherence and long-term outcomes.

By addressing these comments and suggestions, the authors can further strengthen their findings and contribute to a growing body of evidence on the benefits of yoga for managing prehypertension and hypertension.

6. PLOS authors have the option to publish the peer review history of their article (what does this mean? ). If published, this will include your full peer review and any attached files.

**Do you want your identity to be public for this peer review?** For information about this choice, including consent withdrawal, please see our Privacy Policy .

Reviewer #1: **Yes: ** Nagendra Kumar Jain

Reviewer #2: **Yes: ** Dr. Deepeshwar Singh

---

## [Author Response · Author response to Decision Letter 0]

14 Jan 2025

Rebuttal Letter

Dear editors and reviewers,

thank you very much indeed for your efforts while reviewing this manuscript and your helpful comments we addressed accordingly. Changes in the manuscript are highlighted in grey as re-quired.

Editors

Thank you, we did our best and double-checked that our manuscript now meets all PLOS ONE´s style requirements.

“Wiebke Kohl-Heckl is a member of a post-doc scholarship by the Karl und Veronica Carstens Foundation Germany. The work itself did not receive any specific funding”

We included the amended Role of Funder statement in both our cover letter and the online submission.

3. PLOS requires an ORCID iD for the corresponding author in Editorial Manager on papers sub-mitted after December 6th, 2016. Please ensure that you have an ORCID iD and that it is vali-dated in Editorial Manager. To do this, go to ‘Update my Information’ (in the upper left-hand corner of the main menu), and click on the Fetch/Validate link next to the ORCID field. This will take you to the ORCID site and allow you to create a new iD or authenticate a pre-existing iD in Editorial Manager.

Our corresponding author now has an ORCID iD in the updated information.- Https://orcid.org/0009-0001-3049-5089.

4. We notice that your supplementary tables are included in the manuscript file. Please remove them and upload them with the file type 'Supporting Information'. Please ensure that each Sup-porting Information file has a legend listed in the manuscript after the references list.

We now uploaded all supplementary tables in a file type ‘Supporting information’.

5. Please remove your figures from within your manuscript file, leaving only the individual TIFF/EPS image files, uploaded separately. These will be automatically included in the reviewers’ PDF.

We removed all figures from the manuscript file as needed.

6. We notice that your supplementary figures are uploaded with the file type 'Figure'. Please amend the file type to 'Supporting Information'. Please ensure that each Supporting Information file has a legend listed in the manuscript after the references list.

We changed the file type to 'Supporting Information' and included a legend in the manuscript.

7. As required by our policy on Data Availability, please ensure your manuscript or supplemen-tary information includes the following:

A numbered table of all studies identified in the literature search, including those that were ex-cluded from the analyses.

If any of the included studies are unpublished, include a link (URL) to the primary source or de-tailed information about how the content can be accessed.

In the Supporting Information, you can find a table listing all excluded studies and their reason for exclusion, named Supplementary Table S1: Table of excluded studies. All included studies are listed in the table for data extraction Table 1: Study characteristics of eligible studies

If applicable for your analysis, a table showing the completed risk of bias and quality/certainty assessments for each study or outcome. Please ensure this is provided for each domain or pa-rameter assessed. For example, if you used the Cochrane risk-of-bias tool for randomized trials, provide answers to each of the signalling questions for each study. If you used GRADE to assess certainty of evidence, provide judgements about each of the quality of evidence factor. This should be provided for each outcome.

We have added both a table for the Risk of Bias Assessment and Quality of Evidence Assess-ment/Results to our Supporting information file (S6 Table, S7 Table and S8 Table).

Table 1 includes all information drawn for the systematic review and meta-analysis. We assessed all included studies for completeness of data relevant to our analysis. This involved reviewing reported outcomes, sample sizes, and any additional variables required for effect size computa-tion. If certain variables critical for analysis or meta-regression were unavailable, we excluded these variables from that specific analysis. To evaluate the potential impact of missing data on our overall findings, we conducted sensitivity analyses. No substantial differences were found, suggesting that missing data did not significantly bias the result.

We also added a further note to the manuscript regarding the needed information:

“All extracted data are presented in Table 1, which only includes studies that were found eligible, thus provides transparency about missing data. All data were extracted by two authors, Christoph Geiger and Wiebke Kohl-Heckl, between April 5th, 2024 and May 1st, 2024. No further sources but the original articles and their supporting information were used to extract the data needed for the systematic review and meta-analysis. “

Reviewer #2

Overall, the authors have conducted a comprehensive and well-executed systematic review and meta-analysis. The study's strengths include a thorough search strategy, clear inclu-sion/exclusion criteria, and a robust assessment of risk of bias. The findings are generally con-sistent with previous research and provide valuable insights into the potential benefits of yoga for managing prehypertension and hypertension.

Thank you very much for this evaluation of our work! We appreciate it.

Specific Comments:

Heterogeneity: The authors acknowledge the high degree of heterogeneity among the included studies. It would be beneficial to explore potential sources of heterogeneity, such as differences in yoga styles, duration of interventions, and participant characteristics. This could help identify subgroups that may respond differently to yoga.

Thank you for this comment. While we have already included this in our limitations section in the original manuscript as it effects heterogeneity and quality of evidence:

“General limitations are based on methodological reasons, incomplete reporting and a low qual-ity of study performance in general. These limitations include different BP measurement tech-niques, a wide range of BP levels, variation in (control) interventions e.g. passive vs. active con-trol, type of yoga, duration of the intervention period, minutes of yoga applied, as well as medi-cation status, characteristics of study populations and more. “

Further, we also added this to the discussion:

„This heterogeneity of our included studies is based on multiple influencing factors all remain-ing on study design. As summarized in Table 1, some RCTs based their interventions on breath-ing exercises or physical exercises only, while others included multiple aspects of Yoga. As well, studies showed various durations and only some collected follow-up data. Further, gender and age distribution varied.“

Dose-Response Relationship: The authors might consider investigating a dose-response relation-ship between the intensity or duration of yoga interventions and blood pressure reduction. This could provide valuable information for tailoring yoga programs to individual needs and prefer-ences.

Thank you for this suggestion. We have already included this aspect in the original manuscript in the meta-regression. As well, it has been mentioned in the discussion:

“We hypothesized that yoga interventions of longer duration would have larger effects on SBP, DBP and HR, unfortunately our meta-regression did not support our assumption. There was no significant correlation between total duration of yoga interventions and lowering effects on SBP, DBP and HR for studies comparing yoga to usual care. […]”

Quality of Evidence: While the authors note the very low quality of evidence for some of their findings, it's important to discuss the implications of this for clinical practice and future re-search. Are the observed effects still clinically relevant despite the low quality of evidence? What steps can be taken to improve the quality of future studies?

Thank you for marking this aspect. We added it to the discussion:

„Even though we only see low-quality evidence so far based on methodological aspects men-tioned in the limitations´ section, yoga may play an important role in controlling BP. Yoga has the advantage to meet multiple mechanisms for lowering BP, depending on its incorporated aspects such as relaxation, meditation, breathing and physical activation.”

Further, this has been added to the limitations section:

„To be able to further analyze the role of yoga in controlling BP, future studies need to meet comparable criteria, such as 24h-ABPM, more standardized programs and durations. As well, it would be helpful to integrate long follow-up periods to evaluate long-term effects and duration of such.“

Long-Term Effects: Given the chronic nature of hypertension, it would be valuable to explore the long-term effects of yoga on blood pressure management. This could include studies with longer follow-up periods to assess the sustainability of the observed benefits.

Thank you very much for this suggestion. We absolutely see the point and included all available studies that were based on non-single interventions. Unfortunately, there are no studies availa-ble that observe longer-term effects as you can extract from the information given in Table 1. The longest follow-up period performed in any of the available studies was 28 weeks.

Cost-Effectiveness: The authors might consider discussing the cost-effectiveness of yoga as a treatment option for prehypertension and hypertension. This could help inform decision-making by healthcare providers and policymakers.

Thank you for this great suggestion. We included this in our discussion:

„Especially in prehypertensive individuals, where first-line treatment should be based on life-style-modification before including medication, yoga can play an important role. It meets these various aspects of optimizing the autonomous nervous system and includes physical activation, all with the potential of lowering BP. In hypertensive individuals, it can be added as a low-cost therapeutic approach, that can also be performed from home after short period of training or supported by standardized video/online courses, if available. “

Additional Suggestions:

Subgroup analyses: Consider conducting subgroup analyses based on factors such as age, sex, and baseline blood pressure levels. This could help identify populations that may benefit most from yoga.

Thank you for these thoughtful suggestions for subgroup analyses. Based on the available infor-mation of each study included, you can see that the suggested effects are not available as we do not have studies that are only based on e.g., either male or female groups/certain age groups/etc. In our original manuscript, we have performed subgroup analyses, one that com-pares prehypertensive to hypertensive to mixed groups. Further evaluation is not possible based on the information given by the original studies

Mechanism of Action: Discuss potential mechanisms through which yoga may lower blood pres-sure, such as stress reduction, improved cardiovascular fitness, or alterations in the autonomic nervous system.

Thank you very much for this idea. We have added this part to the discussion:

„The mechanisms affecting the BP are various. On the one hand it has stress-reducing effects that are most likely due to a regulation of the function of the autonomic nervous system [13]. Besides, the physical postures, depending on the intensity and duration of hold, may result in an optimized cardiac function and muscular function, if executed regularly. This may lead to an overall improved physical function, facilitating daily life.“

Comparison with Other Lifestyle Interventions: Compare the effects of yoga with other lifestyle interventions, such as exercise, dietary changes, and stress management techniques.

Thank you for this suggestion. As subgroup analysis, we compared active interventions to usual care/control group. Comparing Yoga to other lifestyle interventions would be a very good idea for a future network-meta-analyses, but defeats the idea of this systematic review and meta-analysis.

Patient Preferences: Explore patient preferences for yoga as a treatment option. This could help identify factors that may influence adherence and long-term outcomes.

Thank you for this idea. Unfortunately, neither of the studies mentioned any patient prefer-ences. As this is a very good idea this is a good idea for further research, such as a qualitative study based on patients` interviews, we have added this to our discussion section.

“As well, it is important to evaluate which population may be interested in incorporating yoga into a daily lifestyle for lowering blood pressure. This could for example be achieved by qualita-tive studies such as patient-focused interviews.”

---

## [Decision Letter · Decision Letter 1]

26 Feb 2025

PONE-D-24-24473R1A Systematic Review and Meta-Analysis of Yoga for Arterial HypertensionPLOS ONE

Dear Dr. Geiger,

Thank you for submitting your manuscript to PLOS ONE. After careful consideration, we feel that it has merit but does not fully meet PLOS ONE’s publication criteria as it currently stands. Therefore, we invite you to submit a revised version of the manuscript that addresses the points raised during the review process.

 Please provide a justification for the use of only 3 databases as requested by Reviewer 3.

We look forward to receiving your revised manuscript.

Kind regards,

Emma Campbell, Ph.D

Staff Editor

PLOS ONE

On behalf of 

Bhabesh Deka

Academic Editor

PLOS ONE

Journal Requirements:

Additional Editor Comments:

Authors should consider to include response to minor concerns raised by one of the reviewers in the second revision.

Reviewers' comments:

Reviewer's Responses to Questions

**Comments to the Author**

1. If the authors have adequately addressed your comments raised in a previous round of review and you feel that this manuscript is now acceptable for publication, you may indicate that here to bypass the “Comments to the Author” section, enter your conflict of interest statement in the “Confidential to Editor” section, and submit your "Accept" recommendation.

Reviewer #2: All comments have been addressed

Reviewer #3: (No Response)

2. Is the manuscript technically sound, and do the data support the conclusions?

Reviewer #2: Yes

Reviewer #3: Yes

3. Has the statistical analysis been performed appropriately and rigorously? 

Reviewer #2: Yes

Reviewer #3: Yes

4. Have the authors made all data underlying the findings in their manuscript fully available?

Reviewer #2: Yes

Reviewer #3: Yes

5. Is the manuscript presented in an intelligible fashion and written in standard English?

Reviewer #2: Yes

Reviewer #3: Yes

6. Review Comments to the Author

Reviewer #2: Dear Authors,

The responses to the given comments and the manuscript have been revised satisfactorily. The revisions appropriately address all the provided comments.

Herewith, I approve the manuscript for further processing.

Best regards,

Reviewer #3: It is an excellent review with rigorous approach. However, why only three databases were selected? Other relevant database like EBSCO, AYUSH research portal, Web of Science were not used for article search. An author's justification may be added.

7. PLOS authors have the option to publish the peer review history of their article (what does this mean? ). If published, this will include your full peer review and any attached files.

**Do you want your identity to be public for this peer review?** For information about this choice, including consent withdrawal, please see our Privacy Policy .

Reviewer #2: **Yes: ** Dr. Deepeshwar Singh

Reviewer #3: No

---

## [Author Response · Author response to Decision Letter 1]

1 Apr 2025

Dear editors and reviewers,

thank you very much indeed for your efforts while reviewing this manuscript and your helpful comments we addressed accordingly. As no changes to the manuscript were necessary, we did not upload a new version of it.

Reviewer #2

The responses to the given comments and the manuscript have been revised satisfactorily. The revisions appropriately address all the provided comments.

Herewith, I approve the manuscript for further processing.

Thank you very much for this evaluation of our work! We appreciate it.

Reviewer #3

It is an excellent review with rigorous approach. However, why only three databases were select-ed? Other relevant database like EBSCO, AYUSH research portal, Web of Science were not used for article search. An author's justification may be added.

Thank you very much for this comment. As mentioned in our manuscript, our research approach is based on the Preferred Reporting Items for Systematic Reviews and Meta-Analyses (PRISMA) guidelines and the Cochrane recommendations for systematic reviews. Cochrane therefore gives this specific recommendation to utilize three databases:

“The Cochrane Central Register of Controlled Trials (CENTRAL) and MEDLINE, together with Em-base (if access to Embase is available to the review team), should be searched for all Cochrane Reviews.” (https://training.cochrane.org/handbook/current/chapter-04)

While we decided to utilize Scopus instead of Embase – which was not available to our research team –, Scopus covers about most of the result that can be found via Embase. Scopus additional-ly covers a broader range of other research fields and met more or our targeted parameters.

---

## [Decision Letter · Decision Letter 2]

6 Apr 2025

A Systematic Review and Meta-Analysis of Yoga for Arterial Hypertension

PONE-D-24-24473R2

Dear Dr. Geiger,

We’re pleased to inform you that your manuscript has been judged scientifically suitable for publication and will be formally accepted for publication once it meets all outstanding technical requirements.

Kind regards,

Satish G Patil, PhD

Academic Editor

PLOS ONE

Reviewer's Responses to Questions

**Comments to the Author**

1. If the authors have adequately addressed your comments raised in a previous round of review and you feel that this manuscript is now acceptable for publication, you may indicate that here to bypass the “Comments to the Author” section, enter your conflict of interest statement in the “Confidential to Editor” section, and submit your "Accept" recommendation.

Reviewer #2: All comments have been addressed

Reviewer #3: All comments have been addressed

2. Is the manuscript technically sound, and do the data support the conclusions?

Reviewer #2: Yes

Reviewer #3: Yes

3. Has the statistical analysis been performed appropriately and rigorously? 

Reviewer #2: Yes

Reviewer #3: Yes

4. Have the authors made all data underlying the findings in their manuscript fully available?

Reviewer #2: Yes

Reviewer #3: Yes

5. Is the manuscript presented in an intelligible fashion and written in standard English?

Reviewer #2: Yes

Reviewer #3: Yes

6. Review Comments to the Author

Reviewer #2: Dear Author,

Thank you for providing detailed and satisfactory responses to the comments. The manuscript has now been found suitable and can be considered for further processing.

Best wishes,

Reviewer #3: Author's response to comment is satisfactory. Editorial boards may take decision for accepting the manuscript.

7. PLOS authors have the option to publish the peer review history of their article (what does this mean? ). If published, this will include your full peer review and any attached files.

**Do you want your identity to be public for this peer review?** For information about this choice, including consent withdrawal, please see our Privacy Policy .

Reviewer #2: **Yes: ** Dr. Deepeshwar Singh

Reviewer #3: No

---

## [Editor Report · Acceptance letter]

PONE-D-24-24473R2

PLOS ONE

Dear Dr. Geiger,

I'm pleased to inform you that your manuscript has been deemed suitable for publication in PLOS ONE. Congratulations! Your manuscript is now being handed over to our production team.

Kind regards,

on behalf of

Dr. Satish G Patil

Academic Editor

PLOS ONE